# Arming Immune Cells for Battle: A Brief Journey through the Advancements of T and NK Cell Immunotherapy

**DOI:** 10.3390/cancers13061481

**Published:** 2021-03-23

**Authors:** Philipp Wendel, Lisa Marie Reindl, Tobias Bexte, Leander Künnemeyer, Vinzenz Särchen, Nawid Albinger, Andreas Mackensen, Eva Rettinger, Tobias Bopp, Evelyn Ullrich

**Affiliations:** 1Children’s Hospital, Division for Stem Cell Transplantation, Immunology and Intensive Care Medicine, Goethe-University Frankfurt, 60590 Frankfurt am Main, Germany; philipp.wendel@kgu.de (P.W.); lisamarie.reindl@kgu.de (L.M.R.); tobias.bexte@kgu.de (T.B.); leander.kuennemeyer@kgu.de (L.K.); nawid.albinger@kgu.de (N.A.); eva.rettinger@kgu.de (E.R.); 2Experimental Immunology, Goethe-University Frankfurt, 60590 Frankfurt am Main, Germany; 3Institute for Experimental Cancer Research in Pediatrics, Goethe-University Frankfurt, 60528 Frankfurt am Main, Germany; v.saerchen@kinderkrebsstiftung-frankfurt.de; 4Department of Medicine 5, University Hospital Erlangen, University of Erlangen-Nuremberg, 91054 Erlangen, Germany; Andreas.Mackensen@uk-erlangen.de; 5Institute for Immunology, University Medical Center, Johannes Gutenberg-University Mainz, 55131 Mainz, Germany; boppt@uni-mainz.de; 6Research Center for Immunotherapy (FZI), University Medical Center Mainz, 55131 Mainz, Germany; 7University Cancer Center Mainz, University Medical Center, 55131 Mainz, Germany; 8German Cancer Consortium (DKTK), Partner Site Frankfurt/Mainz, 69120 Heidelberg, Germany; 9Frankfurt Cancer Institute, Goethe-University Frankfurt, 60590 Frankfurt am Main, Germany

**Keywords:** NK cell, T cell, CIK cell, NK-92, cell therapy, immune therapy, CAR-T cell, CAR-NK cell

## Abstract

**Simple Summary:**

This review is intended to provide an overview on the history and recent advances of T cell and natural killer (NK) cell-based immunotherapy. While the thymus was discovered as the origin of T cells in the 1960s, and NK cells were first described in 1975, the clinical application of adoptive cell therapies (ACT) only began in the early 1980s with the first lymphokine activated killer (LAK) cell product for the treatment of cancer patients. Over the past decades, further immunotherapies have been developed, including ACT using cytokine-induced killer (CIK) cells, products based on the NK cell line NK-92 as well as specific T and NK cell preparations. Recent advances have successfully improved the effectiveness of T, NK, CIK or NK-92 cells towards tumor-targeting antigens generated by genetic engineering of the immune cells. Herein, we summarize the promising development of ACT over the past decades in the fight against cancer.

**Abstract:**

The promising development of adoptive immunotherapy over the last four decades has revealed numerous therapeutic approaches in which dedicated immune cells are modified and administered to eliminate malignant cells. Starting in the early 1980s, lymphokine activated killer (LAK) cells were the first ex vivo generated NK cell-enriched products utilized for adoptive immunotherapy. Over the past decades, various immunotherapies have been developed, including cytokine-induced killer (CIK) cells, as a peripheral blood mononuclear cells (PBMCs)-based therapeutic product, the adoptive transfer of specific T and NK cell products, and the NK cell line NK-92. In addition to allogeneic NK cells, NK-92 cell products represent a possible “off-the-shelf” therapeutic concept. Recent approaches have successfully enhanced the specificity and cytotoxicity of T, NK, CIK or NK-92 cells towards tumor-specific or associated target antigens generated by genetic engineering of the immune cells, e.g., to express a chimeric antigen receptor (CAR). Here, we will look into the history and recent developments of T and NK cell-based immunotherapy.

## 1. Characteristics, Target Recognition and Antitumor Functionality of T and NK Cells

Natural killer (NK) cells and T cells are key players of the immune system and utilize several defensive mechanisms to protect the body against viral infections and diseases. Their specific roles as active engagers of possible threats set them apart from the non-engaging immune system, including monocytes and B cells. Despite the fact that both cell types share the purpose of efficiently eliminating pathogen-infected or malignant cells, their target recognition and mode of action varies significantly.

Natural killer (NK) cells were first discovered in mice concurrently by Kiessling [1,2] and Herberman [3,4] in 1975 (Figure 1). The natural occurrence of killer lymphocytes has been described in splenocytes of mice that induced specific cytolytic activity against in vitro-grown mouse Moloney leukemia cells [1,2]. These naturally occurring killer cells were first called “N-cells”, distinct from T cells, B cells and macrophages [3,4,5]. In 1986, Lanier and colleagues proposed that the term “naturally occurring cytotoxicity” should be replaced by “non-major histocompatibility complex (MHC) restricted cytotoxicity” and further termed the respective effector cells natural killer (NK) cells [6]. In the same year, Kärre and Ljunggren described the underlying mechanism of the recognition of malignantly transformed or virus-infected cells by NK cells as the “missing self-hypothesis” [7].

As a part of the innate immune system, NK cells represent a first line defense equipped with a broad repertoire of activating and inhibiting receptors, that allow target-specific recognition and controlled cytotoxic activity without prior stimulation [3,8]. In contrast to T cells, NK cells are able to rely on a highly effective mechanism called “missing self-recognition”. This mechanism allows the discrimination of healthy cells from malignant or infected cells based on the associated loss of MHC class I expression and decrease in suppressive feedback provided by MHC-interaction-based inhibitory receptors, such as killer-immunoglobulin-like receptors (KIRs) and natural killer group two molecules (e.g., NKG2A/B) [9,10,11,12,13,14,15,16,17]. The resulting imbalance in favor of activating signals is further complemented by the stress-associated upregulation of ligands for activating receptors, such as the lysis-triggering natural cytotoxicity receptors (NCRs) NKp30, NKp44 or NKp46 as well as NKG2D, which finally lead to activation of NK cells, modified gene expression and cytolytic activity towards the target cells [18,19,20,21]. Additionally, NK cells utilize receptor-ligand interactions, including the tumor necrosis factor-related apoptosis-inducing ligand (TRAIL) and the Fas ligand (FasL), to promote death receptor-induced apoptosis of the interacting target cell [13]. Furthermore, NK cells provide immune modulatory function by secreting cytokines and chemokines for further regulation of the immune response [22,23,24]. A detailed overview of the killing mechanism of NK cells in context of immunotherapy has recently been reported [13]. On the cellular level, NK cells are defined based on their cluster of differentiation (CD) surface receptor profile as CD56^+^CD3^-^ lymphocytes, and two distinct subpopulations can be classified based on the surface density of CD56 and the low-affinity Fc gamma receptor III (FcγRIII), also known as CD16 [22,25]. The CD56^bright^CD16^dim^ subpopulation and their immune modulatory activity are functionally distinguishable from the CD56^dim^CD16^bright^ population, which represents the NK cell subset with direct cytotoxic capacity [26,27]. Expression of CD16 provides NK cells the ability to synergize with the adaptive immune system by killing antibody-coated (opsonized) target cells through the antibody-dependent cell cytotoxicity (ADCC) pathway, which is not available to T cells [8,22,24,28].

Although the innate immune system can overcome many obstacles on its own, the role of the adaptive immune system is crucial in the second line of defense that leads to acquired immunity.

In the 1960s, Jaques Miller first identified the thymus as the origin of T cells (Figure 1) [29,30,31,32,33]. Until that time, the thymus was considered to be a rudimentary structure and a “graveyard” for lymphocytes [34,35]. By thymus transfer experiments in neonatally thymectomized mice, Miller showed that immune function could be restored and that these mice rejected allogeneic but not donor-derived skin grafts [33], revealing the immunological function of the thymus as the origin of T cells and its relevance in establishing self-tolerance. Subsequent studies demonstrated that mouse lymphocytes can be classified into two distinct lineages, which are either derived from the bone marrow (B cells) or thymus (T cells) [36]. Furthermore, it has been described that T cells undergo mitosis upon antigen stimulation, while B cells show great proficiency at secreting antibodies but only upon interaction with the prior antigen-activated T cells [37,38,39,40]. Based on these findings, the earliest function for T cells was described as that of B cell helpers, leading to the term T helper (TH) cells.

Since the discovery of their potent cytotoxic capabilities, T cells have been the focus of various immunotherapeutic approaches. Their highly specific recognition of malignant or infected cells is based on the interaction of a compatible membrane-bound T cell receptor (TCR) with short peptide motifs related to tumor-associated antigens (TAA) or viral proteins that are presented on the MHC or human leukocyte antigens (HLAs) by antigen presenting cells as well as other nucleated cells [41,42,43]. The resulting mode of action after antigen-interaction is heavily dependent on the T cell subtype.

In the early stages of T cell development, clonal TCRs are formed after rearrangement of the α, β, γ and δ chains, followed by a TCR-mediated positive selection. This process results in maturation of CD4^+^ T helper cells and CD8^+^ cytotoxic T cells [43,44,45,46]. Subsequent negative selection eliminates T cells that recognize self-antigens with high specificity prior to entrance into the circulatory system to prevent potential autoimmune reactions [43,47,48]. The further characterization of mature T cell subsets is based on their phenotype after initial activation by costimulatory receptors as well as secreted cytokines. In general, the heterogenic αβ-TCR T cell population is composed of regulatory CD4^+^ or CD8^+^ T cells (Tregs), which modulate immune responses by several mechanisms (reviewed in Ulges et al. [49]), memory T cells that ensure a fast immune reaction after repeated contact with their specific antigen as well as CD4^+^ TH cells, that contribute to the immune response through additional stimulation of CD8^+^ cytotoxic T lymphocytes (CTLs), which exert the main cytotoxic activity to eliminate infected or malignant cells [17,42,43,50,51,52,53,54]. Further description of the T cell phenotypes can be found in a recently published review article [43]. Concurrently with the maturation of the T cell subtypes, natural killer T cells (NKT cells) develop as a subset of αβ-T cells in the thymus. The unique role of NKT cells is highlighted by their expression of NK-specific receptors, especially CD56 and FcγRIII (CD16) in addition to a specialized αβ-TCR to detect antigens displayed by the monomorphic HLA-like molecule CD1d [55,56,57,58,59]. NKT cells are a highly interesting immune cell population, potentially enabling allogeneic chimeric antigen receptor (CAR)-NKT cell therapies [60]. However, in this review we focus on T and NK cell-based immunotherapies as these are the major players in the currently applied cell therapeutic approaches.

In this context, it is also important to mention that graft-versus-host-disease (GvHD) is a major complication after both stem cell transplantation and donor lymphocyte infusion (DLI), which is highly associated with the alloreactivity mediated by T cells. In contrast, NK and cytokine induced killer (CIK) cells represent attractive targets for immunotherapy without risk of GvHD [61,62]. Although the limited availability of primary NK cells from peripheral blood or umbilical cord blood preparations can be circumvented by optimized expansion protocols, other sources, such as induced pluripotent stem cells (iPSCs) or NK cell lines, have been developed [63]. In particular, the NK-92 cell line is a promising alternative due to its easy expansion and similar characteristics to primary NK cells [64,65]. Notably, NK-92 cells lack several inhibitory receptors, resulting in high levels of activation, but also CD16, which excludes their possible application in ADCC-based therapies [66]. Furthermore, NK-92 cells must be irradiated prior to their application due to the lymphoma origin of this cell line, leading to a reduced in vivo proliferative capacity. The effect of irradiation on the cytotoxic capacity of NK-92 cells is controversially discussed [67,68].

## 2. Cell Therapeutics: The Past, the Present and the Future

### 2.1. A Historical Review of the Development of Cell Therapeutics

#### 2.1.1. Lymphokine-Activated Killer (LAK) Cells

At the beginning of the 1980s, the group of Steven Rosenberg first described a novel approach to generate another type of cytotoxic effector cells for adoptive immunotherapy: lymphokine-activated killer (LAK) cells [69,70]. LAK cells were generated from freshly isolated peripheral blood mononuclear cells (PBMCs) that were cultured in the presence of interleukin-2 (IL-2) and were further characterized by their ability to kill NK-resistant human tumor cells, even though most LAK cells express NK markers (Figure 2) [71]. Today, it is a known fact that LAK cells are composed of a large number of NK and T cells. A huge milestone in the development of LAK-based immunotherapy as well as other immunotherapeutic fields was the isolation of human *IL-2* cDNA from the Jurkat cell line in 1983 [72]. The first studies in murine tumor models revealed that administration of LAK cells alone did not lead to reduced growth of pulmonary metastases, while simultaneous administration of LAK cells and IL-2 resulted in a significant reduction in established pulmonary sarcoma metastases [73,74,75]. In 1985, Steven Rosenberg and colleagues demonstrated the safe administration of LAK cells for patients with metastatic melanoma refractory to standard therapies in a phase I trial [76]. However, in later phase II and III clinical trials comparing the use of IL-2 therapy alone or in combination with LAK cells, no beneficial impact of LAK cells on the patient’s outcome could be observed [76,77]. Due to the low inherent cytotoxic activity of LAK cells, the need for large numbers of cells and the severe side effects that can result from the administration of IL-2, the development of LAK cell-based protocols was not pursued any further, but replaced by more specific immune cell therapies [75].

#### 2.1.2. T Cells

In 1986, Rosenberg and colleagues described the cytotoxic potential of IL-2 expanded tumor-infiltrating lymphocytes (TILs), recognizing tumor-associated antigens from resected tumors in mice [78]. This discovery of the high cytotoxic potency of TILs was followed up in 1988 by the first phase I trial investigating autologous TILs in combination with IL-2-administration [79]. Here, lymphocytes could be successfully isolated from resected melanomas, followed by in vitro expansion of TILs using IL-2 and confirmation of the specific lysis of autologous tumor cells (Figure 2). After initial treatment with chemotherapy, the patients bearing metastatic melanoma were treated by administration of IL-2 and TILs. Although adoptive cell therapy (ACT) using TILs resulted in regression of cancer for several months, the treatment with IL-2 led to toxic side effects [79].

Two years later, Kolb and colleagues reported the first DLI in a phase I trial with three patients suffering from recurrent chronic myeloid leukemia (CML) [80]. Following bone marrow transplantation, lymphocytes from blood of the allogeneic donor were isolated and administered to the patients in combination with interferon α (IFN-α), which resulted in transient cytogenetic remission [80]. Furthermore in 1994, O’Reilly and colleagues achieved complete responses in patients suffering from Epstein-Barr virus (EBV)-associated post-transplant lymphoproliferative diseases (PTLD) after chemotherapy and allogeneic T cell depleted bone marrow transplantation by administration of fresh donor lymphocytes [81]. Based on this idea, prevention of PTLD in addition to tumor regression was demonstrated shortly thereafter [82,83].

In parallel to DLI- and TIL-based immunotherapies, the characterization of TCR-heterodimers recognizing tumor antigens and the development of chimeric antigen receptors (CARs) marked the next steps for tumor-targeting with genetically engineered T lymphocytes (Figure 2) [84,85,86,87,88] (for review also see Sadelain et al. [89]). The introduction of genetically modified immune cells has significantly influenced the immunotherapeutic research for the next decades to come. First, the redirection of T cell specificity by gene transfer of the TCR α/β chain could be successfully demonstrated for transgenic mice in 1986 [90]. However, it took a while until the first reports appeared describing an efficient transfer of melanoma-specific TCRs to human peripheral blood lymphocytes, which led to an increased antigen-specific HLA-A2-restricted CTL response [84,91]. While the introduction of preselected TCR genes into autologous T cells enabled a highly efficient tumor-specific CTL-response, it is important to note that such TCR-mediated antigen recognition is HLA-restricted. Furthermore, generation and optimization of novel, “non-natural” TCRs by structure guided design and affinity maturation allowed the implementation of enhanced TCRs against different tumor entities. However, the HLA-restricted target-recognition remained a hurdle due to MHC-specificity, on-target/off-tumor toxicity and tumor escape mechanisms, such as MHC-downregulation [92,93,94,95].

Therefore, another important milestone was the generation of antigen-specific CTL-hybridoma redirected by a single chain variable fragment (scFv)-based chimeric molecule that acted in a HLA-independent manner [96]. Based on this finding, Eshhar and colleagues developed the first generation of a CAR construct which carried the antigen-binding domain of a Neu/Her2 antibody as an extracellular domain and an intracellular CD3ζ signaling domain to promote the MHC-independent recognition pathway for T cells and an effective elimination of adenocarcinoma cells upon interaction with the target antigen [88]. Although the first-generation CARs had revealed promising results during preclinical assessment, they demonstrated only limited cytotoxic efficacy and persistence in vivo during clinical trials [97]. In order to further improve the functionality of the CAR constructs, a second-generation CAR was equipped with an additional CD28 costimulatory domain in 2002 [98]. Two years later, a second-generation CAR based on the 4-1BB costimulatory domain was published [99]. Both CAR designs proved an increased proliferation capacity and excellent cytotoxic potency of retrovirally transduced CAR-T cells [98,99].These findings promoted the application of CD19-CAR-T cells in the first successful phase I trial for treatment of chronic lymphoblastic leukemia (CLL) in 2011 [100]. Shortly thereafter, CD19-CAR-T cell therapy trials were conducted in three centers for treatment of relapsed childhood and adult acute lymphoblastic leukemia (ALL) and revealed an astonishing complete remission rate [101,102,103,104,105,106,107,108]. 

In the following years, the first third-generation CAR with two costimulatory domains in the intracellular signaling domain of the CAR was designed, along with the clinical trials testing the second-generation CD19-CARs [109]. Furthermore, in 2017, the first implementation of the clustered regularly interspaced short palindromic repeats (CRISPR) -technolgy utilizing the CRISPR-associated endonuclease 9 (Cas9) in a CD19-CAR-T cell product via the *TRAC* locus of T cells was successfully reported [110]. For further review of the history of adoptive T cell therapy see Leon et al. [111].

#### 2.1.3. Natural Killer Cells

Some years after the initial application of LAK cells, a first feasibility trial evaluated the administration of ex vivo purified and expanded autologous NK cells in combination with continuous infusions of IL-2 in patients with metastatic renal cell carcinoma (Figure 2) [112]. Further trials have investigated the transfer of autologous NK cell infusions in combination with IL-2 following conditioning with high-dose chemotherapy [113,114]. Even though autologous NK cell therapy has proven to be feasible, obvious antitumor effects could not be observed. It was hypothesized that the suppression of autologous NK cells might be due to the expression of self-MHC molecules on the surface of tumors, leading to the engagement of inhibitory KIRs [113,114]. Furthermore, IL-2 has been reported to induce the expansion of Tregs, which can either indirectly inhibit NK cell expansion by depriving NK cells of IL-2 or directly inhibit NK cell function in a transforming growth factor (TGF)-β-dependent manner [115,116]. In 2002, Ruggeri et al. showed that KIR ligand-mismatched hematopoietic stem cell transplantation could trigger NK cell alloreactivity [117,118]. These beneficial effects of KIR-mismatched NK cell-mediated alloreactivity could not be consistently confirmed in subsequent studies, possibly due to differences in the transplantation protocols differing in preconditioning regimen or source and dosing of stem cells [119]. In a pilot study, the first allogeneic NK cell products were tested following haploidentical stem cell transplantation (HSCT) [120]. Further clinical trials revealed the safety of the administration of ex vivo purified allogeneic NK cells prior to or after HSCT [121,122,123,124].

In 2005, Jeffrey Miller and colleagues investigated the first allogeneic NK cell therapy in a non-transplant setting and found that a more intensive conditioning regimen consisting of high-dose cyclophosphamide (Cy) and fludarabine (Flu) facilitates NK cell persistence and expansion in vivo, while patients receiving a lower intensity conditioning showed only transient persistence of the administered NK cells [125]. Despite the demonstration of NK cell persistence and proliferation, only moderate clinical responses have been achieved. Further clinical trials have evaluated different NK cell sources as well as different ex vivo expansion and activation protocols [8,126,127]. In 2016, Yang et al. published the results of a first clinical trial in which allogeneic NK cells from unrelated donors were transferred without prior immune suppression for treatment of malignant lymphomas and solid tumors (NCT01212341) [128,129]. Other landmark trials of adoptively transferred NK cells in a non-transplant setting were reported by Iliopoulou et al. [130], Rubnitz et al. [131] (NCT00697671, NCT00187096), Curti et al. [132] (NCT00799799) and Geller et al. [133].

In addition to the administration of autologous or allogeneic NK cell infusions, the safety and feasibility of the adoptive transfer of NK-92 cells after irradiation was assessed in patients with renal cell carcinoma, melanoma and with relapsed or refractory hematological malignancies [134,135,136,137,138]. NK-92 infusions have been proven to be safe with only mild toxicities for treatment of different types of cancer [138]. Based on these beneficial properties, NK-92 cells represent an option for an “off-the-shelf” good manufacturing practice( GMP)-compliant immunotherapeutic product [139,140].

At the same time, protocols were developed which aimed to generate a specialized NK cell population for adoptive immunotherapy. First, the application of tumor-primed NK cells cultivated in the presence of tumor lysates was addressed in patients with acute myeloid leukemia (AML) [141]. Another study addressed the possible antileukemic impact of memory-like natural killer cells, which were induced by pre-activation with IL-12, -15 and -18, for treatment of AML patients. In this first-in-human phase I trial, clinical responses were seen. In addition, proliferation and expansion of memory-like NK cells following adoptive transfer could be monitored [142].

#### 2.1.4. Cytokine-Induced Killer (CIK) Cells

Simultaneously to the first clinical trials investigating the safety and efficacy of autologous NK cell transfer, Schmidt-Wolf et al. described a protocol to generate cytokine-induced killer (CIK) cells and evaluated their antitumor activity in a severe combined immunodeficient (SCID) mouse/human lymphoma model [143]. In contrast to LAK cells, generated exclusively by IL-2 activation, CIK cells were generated from PBMCs that were cultured in the presence of interferon γ (IFN-γ), recombinant IL-2, a monoclonal antibody (mAb) against CD3 and IL-1α using a cross-sectional protocol for 21 days (Figure 2). This procedure promoted a specific induction of T cells with NK cell-like phenotype (T-NK cells) as effector cells, in comparison to the heterogeneous activated lymphocyte population present in LAK cell products. CIK cells can be generated from PBMCs, leukapheresis products, bone marrow or cord blood in the presence of defined cytokines in vitro, resulting in a heterogeneous cell population. This population is characterized by CD3^+^ T cells with or without the acquisition of CD56^+^ and by CD3^-^CD56^+^ NK cells, together combining a highly proliferative and cytotoxic cell population against a broad spectrum of cancers [143,144,145]. The NK-like cytotoxic capacity of CIK cells besides NKp30, DNAM-1, and LFA-1 has been mainly ascribed to NKG2D, an activating NK cell receptor. In 1994, Lu and Negrin compared the antitumor effects of CIK and LAK cells in a SCID mouse model of human lymphoma. In vitro, LAK and CIK cells showed similar cytotoxic potential. In vivo, CIK-treated mice showed a significant prolongation of survival rates compared to LAK-treated mice [146,147]. In 1999, Schmidt-Wolf et al. investigated CIK cells in patients with metastatic renal cancer, colorectal cancer and lymphoma in a first phase I clinical trial [148]. In this trial, patients received autologous CIK cells that were genetically modified to express IL-2 via electroporation. Their administration was proven to be safe and without major side effects, whereas the clinical outcome was moderate. In a first phase I trial in 2005, autologous CIK cells were given to nine patients suffering from advanced Hodgkin’s disease or non-Hodgkin lymphoma (NHL) [149]. The administration of the cells was safe, and only mild toxicities were observed. In a clinical trial in 2006, Jiang et al. investigated the potential benefits of combining chemotherapy and the administration of autologous CIK cells to patients with advanced gastric cancer [150]. Furthermore, it has been demonstrated that CIK cells reached a maximum with regard to the total number and cytotoxic capacity between days 14 and 21 of culture. Further studies investigated the infusion of CIK cells in combination with chemotherapy. Li et al. performed a phase II trial for patients suffering from lung cancer [151], and Niu et al. used cord blood-derived CIK cells in combination with chemotherapy for the treatment of advanced solid malignancies [152].

In 2010, the international registry of CIK cells (IRCC) was set up and identified 11 clinical trials utilizing CIK cells to date. It was retrospectively shown that nearly all trials used autologous CIK cells (10 of 11 studies) and that 384 of 426 patients showed a clinical response. Between 2008 and 2012, 180 patients were enrolled in a randomized, multicenter phase III trial investigating CIK cell therapy in combination with chemotherapy in newly diagnosed glioblastoma patients [153]. Overall, CIK cell infusions were well tolerated with only minor side effects and CIK cell-treated patients showed a significantly higher disease-free survival rate than the untreated control group.

Further attempts to improve the antitumoral activity and to shorten the in vitro expansion period of CIK cells included the addition of IL-15 on day four of culture [154]. After 10 days, IL-15-stimulated CIK cells showed enhanced killing of AML and lymphoma cell lines as well as primary acute myeloid and lymphoblastic leukemia cells in vitro and to a certain extent improved efficacy of CIK cell therapy compared to conventional DLI in patients with relapsing hematological malignancies after allogeneic HSCT [155].

In 2020, Zhang and Schmidt-Wolf provided a 10-year update of the international registry on CIK cells in cancer immunotherapy, and identified 106 clinical trials including 10,255 patients, of whom 4889 patients with more than 30 tumor entities were treated with CIK cells alone or in combination with standard or novel therapy approaches. In addition to assessing the short-term benefits, they reviewed nine studies that reported a significantly improved 5-year survival rate, and another 27 trials demonstrating improved median progression-free and overall survival rates after CIK cell therapy, which underlines the promising antitumoral activity of CIK cells [156]. Furthermore, adoptive transfer of allogeneic CIK cells confers other advantages, such as the reduced risk of causing GvHD in comparison to T cell therapies, due to limited proliferative capacity and release of cytokines by CIK cells, such as IFN-γ, and lack of further severe side effects, which may protect against GvHD [156,157,158].

### 2.2. Recent Clinical Studies of Immune Cell Therapy

A multitude of clinical trials using adoptive NK and T cell therapies, in which autologous, allogeneic and immortalized cells are applied to fight various types of cancers, are currently being conducted worldwide [159].

One strategy constitutes the additional blockade of inhibitory receptors or receptor ligands, such as checkpoint inhibitors, on tumor and immune cells with mAbs [160,161,162]. Although several clinical trials have been investigating the effects of checkpoint inhibitors on NK and T cells, this novel strategy has yet to deliver convincing results [163,164].

Another approach is the additional treatment of patients with NK or T cell-activating cytokines in combination with an adoptive cell transfer to promote immune cell-activation, cytotoxicity and expansion. For NK cell-based therapies, IL-2, -12, -15, -18 and -21 have been shown to increase NK cell proliferation and cytotoxicity in preclinical studies [159].

A third strategy to increase efficacy of adoptively transferred NK or T cells is the sensitization of tumor cells towards cytotoxic immune cells by certain standard-of-care therapies. This was shown for chemotherapies, endocrine deprivation, sublethal doses of irradiation and various cytotoxic small molecules [163].

For adoptive T cells, over 50 active studies are currently listed on clinicaltrials.gov (accessed on 7 October 2020). Indications range from hematological malignancies such as multiple myeloma or B cell lymphomas to solid tumors such as melanoma, breast cancer and colorectal cancer. Furthermore, in some trials, adoptive T cells are administered to patients suffering from virus-induced diseases such as cytomegalovirus disease (NCT02982902, NCT03594981, and NCT04056533), which had already generated promising results in the past [165,166], or coronavirus disease 2019 (COVID-19) (NCT04457726, NCT04578210 and NCT04351659).

In the field of CAR-modified T cells, hundreds (over 600) of clinical trials are currently being conducted worldwide, which mostly target hematological diseases and most recently also solid tumors. Results from clinical trials led to the approval of Kymriah^®^ (Novartis Pharmaceuticals Corporation, Basel, Switzerland) and Yescarta^®^ (Kite Pharma, Incoporated, Santa Monica, CA, USA) in 2017 and Tecartus™ (Kite Pharma, Incorporated, Santa Monica, CA, USA) in 2020 by the U.S. Food and Drug Administration (FDA). Among these three products, Kymriah and Yescarta were also approved by the European Medicines Agency (EMA) in 2018 [167,168,169]. Kymriah was approved for relapsed or refractory (r/r) B cell acute lymphoblastic leukemia (B-ALL) and r/r large B cell lymphomas [170], while Yescarta was approved for r/r diffuse large B cell lymphoma (DLBCL) as well as primary mediastinal large B cell lymphoma (PMBCL) [169,171], and Tecartus was approved for r/r mantle cell lymphoma (MCL) [172].

On clinicaltrials.gov (accessed on 07 October 2020), no recruiting phase I/II trials using LAK cells are listed at the moment. Since previous clinical studies have already shown that CIK cells are safe, even in a HLA-mismatch setting, and can significantly increase the survival rates of cancer patients, over 35 clinical trials using CIK cells are currently being conducted worldwide [104,148]. Indications include multiple types of cancers, including NHL, acute leukemia, colorectal cancer, liver cancer, breast cancer, melanoma and other types of solid tumors. In accordance with the striking success of CD19-redirected CAR-T cells [100,101,102,103,104,105,106,107,108], CIK cells were engineered to express chimeric receptors, such as CD19-CARs to enhance their cytotoxic activity [173]. An interim update on the phase I/II clinical trial conducted in Monza and Bergamo, Italy, reported promising clinical response, safety with only grade 1 and 2 cytokine release syndrome (CRS) and persistence of CD19-CAR-CIK cells (ASH, 16 December 2020). In a recent study, CIK precursors were genetically modified to express CARs against the tumor-promoting antigen CD44v6 for treatment of high-grade soft-tissue sarcoma [174].

In the context of cell therapeutic approaches using primary NK cells, currently more than 30 active clinical trials are listed on clinicaltrials.gov for adoptive/allogeneic NK cell therapy as reported earlier [8]. Although these studies mainly cover participants suffering from various types of cancers, two studies for the treatment of COVID-19 patients are enlisted as well (NCT04344548 and NCT04365101). Both COVID-19 studies are based on the findings that a reduction of circulating cells and/or an exhausted phenotype of NK cells can be observed in the context of the COVID-19 disease [175].

Moreover, trials of NK cells in combination with immune checkpoint inhibitors (ICIs) have been conducted, which have included the usage of the PD-1 antibody pembrolizumab (NCT03937895). Furthermore, one trial was carried out in which allogeneic NK cells were administered in combination with the humanized GD2 antibody Hu3F8 to treat high-risk neuroblastoma (NCT02650648). In an earlier phase I study, adoptive NK cells in combination with the murine GD2 antibody m3F8 were shown to be safe and possess anti-neuroblastoma activity at higher cell doses (NCT00877110) [176].

Overall, in the majority of studies, patients receive adjuvant or first-line chemotherapy prior to NK cell administration, and in some trials, IL-2 is additionally administered (NCT01857934, NCT04347616, NCT02650648, NCT01823198, and NCT01700946). Earlier studies showed that IL-2 administration can cause durable antitumor activity but also increase the frequency of circulating regulatory T cells and can induce severe side effects such as pneumonitis [177,178]. In addition, adoptive NK cells were tested together with the IL-15 superagonist ALT-803, which has already been demonstrated to promote NK cell function in vitro and in vivo in patients suffering from acquired immune deficiency syndrome (AIDS) (NCT03899480) [179,180].

Following the success of CAR-T cell trials, the first CAR-NK cells tested in clinical studies were directed against CD19 for treatment of B-lineage acute lymphoblastic leukemia (ALL) (NCT00995137, NCT01974479). Both trials used a second-generation CAR and expanded NK cells in presence of irradiated and genetically modified K562 feeder cells and demonstrated safety and feasibility [8]. Another trial investigating CD19-CAR-NK cells at the M.D. Anderson Cancer Center (Houston, TX, USA) analyzed umbilical cord blood (UCB)-derived NK cells as a source for the adoptive transfer of CAR-NK cells for patients suffering from CD19-expressing B cell malignancies (NCT03056339). Out of the first 11 patients treated in this study, eight patients showed a clinical response, of whom seven had complete remission. A special feature of this trial was the incorporation of a suicide gene (inducible caspase 9) and a gene to ectopically produce IL-15 to improve the lifespan and persistence of NK cells in the patient [181]. Clinical trials with primary CAR-NK cells are much less common, with currently only 10 active trials worldwide. In addition to hematological malignancies, the effect of CAR-NK cells on other diseases, such as epithelial ovarian cancer (NCT03692637), pancreatic cancer (NCT03941457) or COVID-19 (NCT04324996) is under investigation. Although CAR-NK cells demonstrate promising results in previous and ongoing clinical trials, no CAR-NK cell product has received market approval yet [8,181].

The first in-human trial of CAR-expressing NK-92 cells for treatment of patients with relapsed or refractory AML was conducted by Tang and colleagues (NCT02944162). In their trial, a third-generation CD33-CAR construct was used to generate CD33-CAR-NK-92 cells. The administration of pre-irradiated CD33-CAR-NK-92 cells was reported as safe and the engineered NK-92 cells could be detected up to one week post-infusion, whereas clear clinical efficacy could not be demonstrated [65].

Currently, 12 ongoing clinical trials are in the process of evaluating NK-92 cells for treatment of various types of cancers, such as multiple myeloma, glioblastoma and pancreatic cancer. In some of these trials, genetically modified NK-92 cells expressing CD16 and intracellular endogenous IL-2 (haNK cells) are being deployed [182], in others, NK-92 cells are being administered in combination with cytokines or highly potent super-agonists, such as IL-2/IL-15Rα or IL-15/IL-15Rα (ALT-803) [159,183]. Furthermore, NK-92 cells are being evaluated in combinatorial treatment approaches with a multitude of other drugs, such as checkpoint inhibitors [163]. At the moment, two studies are conducted which utilize unmodified NK-92 cells (NCT02465957 and NCT02727803) as well as two studies which deploy genetically modified CAR-NK-92 cells (NCT03940833 and NCT03383978) [8].

### 2.3. Hurdles and Improvements for Immune Cell Therapy

The peculiarity of tumor reactive lymphocytes that co-reside in vivo together with cancer cells and do not recognize and attack the malignant cells was termed the *Hellström paradox* [184]. Although impressive progress has been made to tackle inefficient antitumor immunity, many hurdles are still present in immunotherapy, such as immunosuppression by tumor immune escape mechanisms [185] and GvHD [186].

An interesting approach is the combination of the genetic modification of immune cells and immune checkpoint inhibitors. With the aim to minimize the systemic effect of immune checkpoint blockade (ICB) therapy, CAR-T cells were further modified to express and secrete an anti-PD-1 single-chain variable fragment for targeted and augmented CAR-T cell activity [187]. The promising synergistic effect of CAR-T cell therapy and ICB, which may help to overcome T cell exhaustion, may also come at a price. Cases were found in which, post CAR-T cell therapy, treatment with pembrolizumab had led to immune-related adverse reactions [188]. For a conclusive evaluation of the safety of ICB post CAR-T cell therapy, more follow-up studies and case reports need to be analyzed. In order to explore the combination of CAR-T cell therapy with ICB, several other strategies have been introduced. CRISPR/Cas9 gene editing can be used to disrupt the inhibitory PD-1 axis [189,190]. Furthermore, CRISPR/Cas9 can be used to direct the CAR construct to a known and non-random genomic location to disrupt the endogenous TCR locus [110].

Furthermore, altered levels of pro- and anti-apoptotic proteins, such as B cell lymphoma 2 (BCL2) proteins and inhibitor of apoptosis proteins (IAP) [191,192] as well as downregulation of MHC proteins [193], recruitment of Tregs and myeloid-derived suppressor cells (MDSC) [194,195] lead to the development of an immunosuppressive tumor microenvironment (TME) [196,197]. Tumor cells do not only react by upregulation of inhibitory ligands, but also by downregulation or by shedding of activating ligands necessary for activation of cytotoxic lymphocytes [198]. A promising idea is the inhibition of NKG2D-ligand shedding in combination with the development of NKG2D-CAR-NK cells [11,199]. Advantages of CAR-NK cells in immunotherapy are reviewed in Reindl et al. [8].

Although CAR-T cells specific for CD19 showed high potential antitumor efficacy against relapsed and refractory B-ALL in clinical trials, these therapies are limited to autologous settings even when they are HLA-matched. Furthermore, generation of sufficient numbers of CAR-T cells for patients suffering from severe lymphopenia is impractical. T cell application can be accompanied by adverse side effects such as CRS, GvHD and on-target off-tumor toxicity. An alternative method of cytotoxic CAR-T cell generation is the use of regulatory T cells. Imura and colleagues showed that by generating CD19-CAR-Tregs, the target cells could be suppressed and not eliminated, which diminished the onset of GvHD in mice [200].

Furthermore, researchers have developed different safety strategies to overcome such hurdles in clinical CAR-T cell treatments in addition to investigating new treatments using allogeneic NK cells. One very well-characterized suicide gene is herpes simplex virus-thymidine kinase (HSV-tk) [201]. It is widely used in combination with ganciclovir (GCV) [202,203]. It is very often used to eliminate GvHD after donor lymphocyte therapy post-HSCT [204,205]. However, preclinical data demonstrated that anti-CD44v6-CAR-T cells equipped with the HSV-tk suicide gene could be eliminated after exposure to GCV in AML and multiple myeloma [206]. As T cell death is based on interference with DNA synthesis, gradual progress will need to be made, as it takes time to eliminate CAR-T cells by HSV-tk [207]. The inducible safety switch suicide gene caspase 9 (iCasp9) showed a much faster response to apoptosis of CAR-T cells after conditional administration of AP1903. In several clinical trials using iCasp9/ΔCD19 T cells (NCT00710892) or similar constructs with iCasp9 (NCT01494103), GvHD symptoms were alleviated within 24 h, and iCasp9 T cells were eliminated very rapidly, leading to rapid remission of GvHD and CRS [208,209].

Further advantages were shown by using CARs co-expressing CD20 or truncated epidermal growth factor receptor (EGFRt). These CARs can be effectively eliminated with the therapeutic monoclonal antibodies rituximab or cetuximab [210,211]. In particular, the EGFRt-rituximab based suicide switch has been applied effectively in numerous clinical trials [212].

Regarding the ongoing investigation to resolve tumor immune evasion mechanisms (TIEM), new approaches divide the traditional CAR into two complementary parts using two distinct antigen-recognition domains (tandem CAR) or two single CAR constructs (dual CAR) [213] recognizing two different antigens, resulting in reduced antigen escape and “on-target off-tumor” toxicity and enhanced tumor specificity [214,215,216]. New constructs have been generated for the modulation of the tumor microenvironment by additional inducible cytokine release after T cell activation, which have been incorporated into so-called T cells redirected for universal cytokine killing (TRUCKs) [217]. With the aim to further optimize the antigen response and remodulate local microenvironments, novel designed synthetic notch (synNotch) receptors release intracellular transcription factors upon activation [215].

Additional split CAR technologies, such as the Adapter CAR technology (AdCAR) have been combined with other recent approaches. The AdCAR utilizes biotinylated mAbs as adapter molecules with a universal CAR system to regulate immune effector cell function. This allows not only universal but furthermore a combinatorial immune targeting [218,219].

Classical CAR-NK cell products show difficulties in migrating to the required location. One approach, which is also applicable to solid tumors, is the generation of transgenetically augmented CAR-NK cells (TRACKs). As a first proof-of-concept, CXCR4 was introduced into CD19-CAR-NK cells to increase their homing capacity towards bone marrow stromal cells [220].

In order to further circumvent T cell-related side effects, the development of allogeneic immune cell products such as primary NK cells or the NK-92 cell line is a highly interesting and promising approach. In a very recently published first clinical trial using CD19-CAR-NK cell treatment, patients showed response rates of 73%, and no GvHD, CRS or neurotoxicity has been reported to date [181]. Although gene modification of immune cells through the use of viral-vectors is feasible, the GMP-grade production of viral-vectors for clinical application remains cost- and time-intensive with high safety standard requirements. As an alternative engineering system for CAR-immune cell products, non-viral alternatives gained increasing attention [8]. CD19-CAR-T cells with the Sleeping Beauty (SB) transposon/transposase system were safe, and no acute or late toxicities or GvHD were observed [221]. Compared with viral transduction, the SB method showed a significantly higher genomic integration profile using minicircle (MC) and SB mRNA compared to lentiviral integration in CD19-CAR-T cells [222]. Lower costs and reduced regulatory demands offer clear advantages for production of non-viral vector-based cell therapeutics under GMP, and successful clinical application has already been reported for CD19-CAR-CIK cells [223,224]. Furthermore, a SB-engineered signaling lymphocytic activation molecule (SLAM) family member 7 (SLAMF7)-specific T cell product for treatment of patients suffering from multiple myeloma is currently under investigation in a phase I/II clinical trial CARAMBA-1 (NCT04499339; EU Horizon 2020 project). Additionally, CIK cells can be CAR-redirected through lentiviral and non-viral transposon system [224], and can exert relevant ADCC upon incubation with clinical-grade mAbs due to CD16 expression [225,226].

In the NK cell field, recently, the transfection of a CAR-containing *PiggyBac* transposon vector in combination with transposase DNA for the generation of CAR-iPSC-NK cells was reported [227], and the development of SB-engineered CAR-NK cell products is also currently under development and showed high cytotoxic efficacy in first preclinical evaluation (unpublished data from the Ullrich group, presented at the 47th EBMT Annual Meeting 2021).

## 3. Summary and Conclusions

The ever-evolving development of immune cell-based cancer therapy has been characterized by the generation of novel immune cell products, the constant discovery of new cancer-specific antigens and, furthermore, the genetic modification of immune cells to enhance and direct their functionality in vivo.

Although early in vitro experiments suggested LAK cells as a promising immunotherapeutic approach, the beneficial impact in clinical trials fell short of expectations [71,75,76,77]. Nonetheless, the insights into the possibilities of ACT for the treatment of cancer gave rise to various immune cell products shortly afterwards.

Further strategies were the application of DLIs, which especially in combination with IFN-α, induced some tumor regression in addition to the prevention of PTLD [80,81,82,83]. However, a prominent risk of GvHD after infusion of immune cell products containing allogenic T cells remained.

An alternative approach involved in vitro expanded, autologous TILs recognizing tumor antigens from resected tumors. While administration of TILs with IL-2 demonstrated the specific and high cytotoxic potential of the immune cell product against the designated tumor, the application of IL-2 caused toxic side effects [78,79].

In parallel, the characterization of TCR-heterodimers recognizing tumor antigens and subsequent engineering of (non-natural) TCRs opened up the possibility to generate genetically engineered T lymphocytes redirected towards specific cancer entities [84,85,86,87,88]. In this context, it is important to note, that the HLA-restricted target-recognition through the TCR remained a prominent hurdle for T cell-based therapies due to prominent tumor escape mechanisms, such as MHC downregulation [84,90,91,92,93,94,95].

In order to provide an effective and HLA-independent tumor-recognition mechanism genetically engineered T cells were equipped with CARs, that enabled a highly specific antigen recognition utilizing scFvs instead of TCRs. Although the initial CAR generation failed to provide the desired in vivo response, later generations improved the CAR functionality, demonstrated excellent cytotoxic potency, and had long persistence in vivo. Furthermore, the development gave rise to highly-efficient expansion protocols, which eventually led to approval of multiple CAR-T cell products by the FDA and EMA [96,97,98,99,100,101,102,103,104,105,106,107,108,109,167,168,169,170,171,172]. However, the limited availability of tumor-exclusive antigens for certain tumor entities remains a hurdle for CAR-based immune cell products.

Based on the promising antitumoral activity after allogeneic ACT compared to T-cell-exclusive products, CIK cells emerged as a promising immunotherapeutic strategy for treatment of a variety of cancers [149,150,153,156,157,158]. Following the clinical success of CAR-T cell products, further studies harbored the potential of CAR-based immunotherapy to enhance the cytotoxic activity of CIK cells against specific tumor antigens [173,174].

Furthermore, the development of immune cell products based on primary NK cells or the NK-92 cell line combine the highly effective naïve cytotoxic capability of NK cells with a low risk of causing GvHD and other T cell-related side effects in an allogeneic setting, for example due to their distinct cytokine profile [121,122,123,124,134,135,136,137,138,181,228]. This opens up the possibility to offer effective, allogeneic “off-the-shelf” immune cell products for cancer treatment [139,140].

In contrast to T cell-products which can utilize a high clonal expansion and persisting memory T cell response after treatment, primary NK cells have a lowered proliferative capacity and persistence in vivo due to their cytokine-dependency [181]. This limited persistence is even more prominent for the NK-92 cell line, which has to be irradiated prior to application, resulting in a hold of cell proliferation [138].

To counteract these limitations, genetic engineering of NK cells, such as the implementation of CARs combined with the expression of endogenous cytokines, offers a promising opportunity to redirect the high cytotoxic potential towards specific tumor antigens, utilize the naïve cytotoxicity and achieve a long-term persistence in vivo [8,181].

In conclusion, the developments in cell-based immunotherapy for the last four decades enabled the administration of more precise cell therapeutic therapies tailored to various cancer subtypes and opened up promising opportunities for the development of novel immunotherapeutic concepts for the fight against cancer.

## Figures and Tables

**Figure 1 cancers-13-01481-f001:**
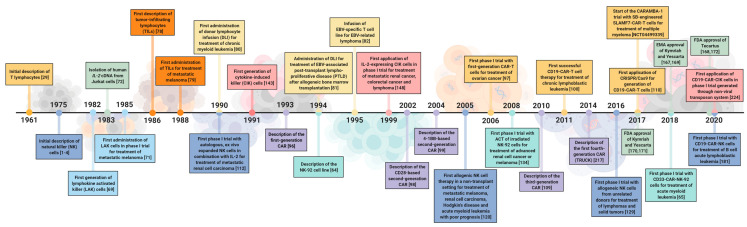
The history of cell-based immunotherapies. Important discoveries, key clinical trials and highlights which were prominent in the last decades of cell-based immunotherapies are illustrated. ACT = adoptive cell therapy; CAR = chimeric antigen receptor; CD = cluster of differentiation; CRISPR = clustered regularly interspaced short palindromic repeats; Cas9 = CRISPR-associated endonuclease 9; EBV = Epstein–Barr virus; EMA = European Medicines Agency; FDA = U.S. Food and Drug Administration; IL-2= interleukin 2; SB = Sleeping Beauty transposon/transposase system; TRUCK = T cells redirected for universal cytokine killing.

**Figure 2 cancers-13-01481-f002:**
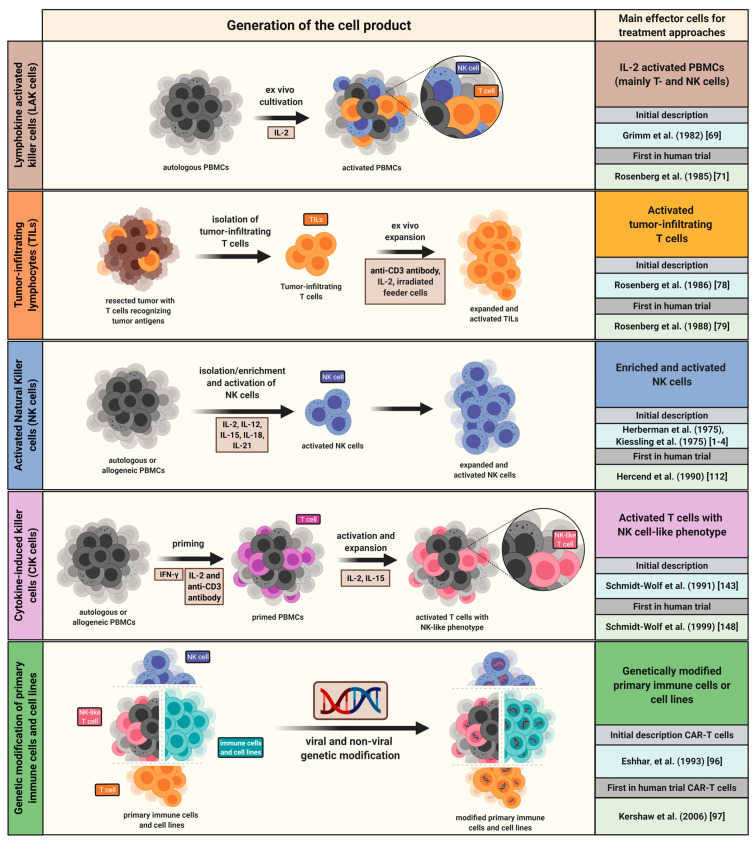
Generation of immune cell products for application in cancer therapy. Overview of exemplary cell sources and procedures for the generation of representative immune cell products. Additionally, time points of discovery and initial application in clinical trials are indicated. Alternative cell sources are possible for specific immune cell products. INF-γ = Interferon-γ; PBMC = peripheral blood mononuclear cells.

## Data Availability

Not applicable.

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
