# Peer review of "Arming Immune Cells for Battle: A Brief Journey through the Advancements of T and NK Cell Immunotherapy"

_cancers, 2021, doi:10.3390/cancers13061481_

Round 1

Reviewer 1 Report

This is a very nice and up-to-date review on the various strategies of NK- and T-cell based immunotherapy of cancer, with an interesting historical perspective. I have few minor comments which could help further improving the manuscript.

It would be nice to list the advantages and drawbacks of each of the approaches discussed, including feasibility. That could be added in the etxt where it is missing, and summarized in Fig. 2 right panel. For example, the fact that only few target antigens are known for CAR therapy is not discussed. Also, it would be nice to implement a paragraph on TCR engineered T cells and discuss the risk and benefit of “non-natural” TCRs (eg. affinity matured).

Regarding NK-based cell products, the authors mention that clinical benefit has been very limited so far; it is not very clear how the new strategies are going to improve this. Also, is there any issue with alloreaction against NK-92 or allogenic NK-cells?

It is not clear for the non-expert what the exact difference between CIK and LAK cells is, please add.

Other minor comments:

Figure 1 shows milestones of adoptive transfer therapies, so the title needs to be more precise, since all immunotherapies are not treated in the review

Fig2, 1st strategy (LAKs). “PBMCs from an autologous blood donor”: I assume the authors mean “autologous PBMCs”?

2nd strategy (TILs): not only TAAs, but also some mutations or further tumor specific antigens might be recognized; I would suggest to correct for “tumor antigens”

It would be nice to add the exact (or the range) of IL-2 doses instead of just “high” which is a bit vague (see lines 148, 173, 189); same comment for the number of infused NK92 cells (line 278).

The quality of the figures is not very high, will this be improved for the final version?

Author Response

Reviewer #1 (Remarks to the Author):

Reviewer #1 comment 1: This is a very nice and up-to-date review on the various strategies of NK- and T-cell based immunotherapy of cancer, with an interesting historical perspective. I have few minor comments which could help further improving the manuscript. It would be nice to list the advantages and drawbacks of each of the approaches discussed, including feasibility. That could be added in the text where it is missing, and summarized in Fig. 2 right panel. For example, the fact that only few target antigens are known for CAR therapy is not discussed. Also, it would be nice to implement a paragraph on TCR engineered T cells and discuss the risk and benefit of “non-natural” TCRs (eg. affinity matured).

Reply 1.1: We thank the reviewer for this thoughtful comment. We included a summary of the benefits and also limitations of immune cell-based therapies in our new “Summary and conclusion” (lines 586-642). Due to space limitations and for visual clarity we decided against the description of advantages and disadvantages in Figure 2, but we aimed to utilize the compact format of our conclusion for this comparison. Furthermore, we want to thank the reviewer for the suggestion to describe the limitation of antigen availability for certain cancer types. We included a short statement the conclusion (lines 616-617, underlined in the following text).

“The ever-evolving development of immune cell-based cancer therapy has been characterized by the generation of novel immune cell products, the constant discovery of new cancer-specific antigens and furthermore the genetic modification of immune cells to enhance and direct their functionality in vivo.      
Although early in vitro experiments suggested LAK cells as a promising immunotherapeutic approach, the beneficial impact in clinical trials fell short of expectations [71,75–77]. Nonetheless, the insights into the possibilities of ACT for the treatment of cancer gave rise to various immune cell products shortly afterwards.
Further strategies were the application of DLIs, which especially in combination with IFN-α, induced some tumor regression in addition to the prevention of PTLD [80–83]. However, a prominent risk of GvHD after infusion of immune cell products containing allogenic T cells remained.
An alternative approach involved in vitro expanded, autologous TILs recognizing tumor antigens from resected tumors. While administration of TILs with IL-2 demonstrated the specific and high cytotoxic potential of the immune cell product against the designated tumor, the application of IL-2 caused toxic side effects [78,79].    
In parallel, the characterization of TCR-heterodimers recognizing tumor antigens and subsequent engineering of (non-natural) TCRs opened up the possibility to generate genetically engineered T lymphocytes redirected towards specific cancer entities [84–88]. In this context, it is important to note, that the HLA-restricted target-recognition through the TCR remained a prominent hurdle for T cell-based therapies due to prominent tumor escape mechanisms, such as MHC downregulation [84,90–95]. 
In order to provide an effective and HLA-independent tumor-recognition mechanism genetically engineered T cells were equipped with CARs, that enabled a highly specific antigen recognition utilizing scFv’s instead of TCRs. Although the initial CAR generation failed to provide the desired in vivo response, later generations improved the CAR functionality, demonstrated excellent cytotoxic potency and long persistence in vivo. Furthermore, the development gave rise to highly-efficient expansion protocols, which eventually led to approval of multiple CAR-T cell products by the FDA and EMA [96–109,167–172]. However, the limited availability of tumor-exclusive antigens for certain tumor entities remains a hurdle for CAR-based immune cell products.
Based on the promising antitumoral activity after allogeneic ACT compared to T-cell-exclusive products, CIK cells emerged as a promising immunotherapeutic strategy for treatment of a variety of cancers [149,150,153,156–158]. Following the clinical success of CAR-T cell products, further studies harbored the potential of CAR-based immunotherapy to enhance the cytotoxic activity of CIK cells against specific tumor antigens [173,174].       
Furthermore, the development of immune cell products based on primary NK cells or the NK-92 cell line combine the highly effective naïve cytotoxic capability of NK cells with a low risk of causing GvHD and other T cell-related side effects in an allogeneic setting, for example due to their distinct cytokine profile [121–124,134–138,181,228]. This opens up the possibility to offer effective, allogeneic “off-the-shelf” immune cell products for cancer treatment [139,140].
In contrast to T cell-products which can utilize a high clonal expansion and persisting memory T cell response after treatment, primary NK cells have a lowered proliferative capacity and persistence in vivo due to their cytokine-dependency [181]. This limited persistence is even more prominent for the NK-92 cell line, which has to be irradiated prior to application, resulting in a hold of cell proliferation [138].   
To counteract these limitations, genetical engineering of NK cells, such as the implementation of CARs combined with the expression of endogenous cytokines, offers a promising opportunity to redirect the high cytotoxic potential towards specific tumor antigens, utilize the naïve cytotoxicity and achieve a long-term persistence in vivo [8,181].
In conclusion, the developments in cell-based immunotherapy for the last four decades enabled the administration of more precise cell therapeutic therapies tailored to various cancer subtypes and opened up promising opportunities for the development of novel immunotherapeutic concepts for the fight against cancer.”

Reply 1.2: We thank the reviewer for this excellent suggestion and added a paragraph on “non-natural” TCRs in chapter “2.1.2. T cells” (lines 221-226).

“Furthermore, generation and optimization of novel, “non-natural” TCRs by structure guided design and affinity maturation allowed the implementation of enhanced TCRs against different tumor entities. Nevertheless, the HLA-restricted target-recognition remained a hurdle due to MHC-specificity, on-target/off-tumor toxicity and tumor escape mechanisms, such as MHC- downregulation [92–95].”

Reviewer #1 comment 2: Regarding NK-based cell products, the authors mention that clinical benefit has been very limited so far; it is not very clear how the new strategies are going to improve this.

Reply 2: We thank the reviewer for this thoughtful comment. We tried to guide the reader trough the development in NK-cell based immunotherapies in the past and also highlight their drawbacks and limitations. Chronologically speaking, the first NK cell products demonstrated only moderate results, but current approaches with primary (CAR-)NK cells applied in an allogeneic setting achieved highly promising result (e.g., Liu et al. 2020) in addition to the low risk of causing GvHD. We included a summary of the benefits and also limitations of NK cell-based immunotherapies in the summary and conclusion. For further reading we have summarized current NK cell-based immunotherapy in Reindl et al. 2020. In fact, there are more than 200 ongoing clinical trials (including NK-92) investigating NK cell-based immunotherapies. For additional clarity we added the follow sentence on the current state of CAR-NK cell approval in lines 453-455.

“Although CAR-NK cells demonstrate promising results in previous and ongoing clinical trials, no CAR-NK cell product has received market approval yet [8,182].”

Reviewer #1 comment 3: Also, is there any issue with alloreaction against NK-92 or allogenic NK-cells?

Reply 3: We thank the reviewer for this remark. To our knowledge, no adverse alloreactions against NK-92 or allogeneic NK cells have currently been described. However, there is a risk of an immunogenic reaction against genetically modified immune cell products, such as CAR-T- or CAR-NK, after administration, as outlined in “Gorovits & Eugen "Immunogenicity of chimeric antigen receptor T-cell therapeutics." BioDrugs (2019)” for CAR-T cells.

Reviewer #1 comment 4: It is not clear for the non-expert what the exact difference between CIK and LAK cells is, please add.

Reply 4.1: We thank the reviewer for this thoughtful comment. After revision of our paragraph on LAK cells we corrected the description of the main effector cells in line 167 from “T-NK cells” to “T cells”.

“Today, it is a known fact that LAK cells are composed of a large number of NK- and T cells.”

Reply 4.2: For the chapter “2.1.4. Cytokine-induced killer (CIK) cells” we added a short comparison of CIK with the former described LAK cell preparations (lines 307-313).

“In contrast to LAK cells, generated exclusively by IL-2 activation, CIK cells were generated from PBMCs that were cultured in the presence of IFN-g, recombinant IL-2, a monoclonal antibody (mAb) against CD3 and IL-1a using a cross-sectional protocol for 21 days (Figure 2). This procedure promoted a specific induction of T cells with NK cell-like phenotype as main effector cells, in comparison to the heterogeneous activated lymphocyte population present in LAK cell products.”

Other minor comments by Reviewer #1:

Reviewer #1 comment 5: Figure 1 shows milestones of adoptive transfer therapies, so the title needs to be more precise, since all immunotherapies are not treated in the review

Reply 5: We fully agree with the reviewer and have changed the title and description of Figure 1 in lines 155-156 towards immunotherapies based on ACTs.

“The history of cell-based immunotherapies. Important discoveries, key clinical trials and highlights which were prominent in the last decades of cell-based immunotherapies are illustrated. This figure has been created using BioRender.com.”

Reviewer #1 comment 6: Fig2, 1st strategy (LAKs). “PBMCs from an autologous blood donor”: I assume the authors mean “autologous PBMCs”?

Reply 6: We thank the reviewer for this important remark and corrected it accordingly in Figure 2 to “autologous PBMCs”. In addition, we adapted the other descriptions of PBMCs in Figure 2.           

Reviewer #1 comment 7: 2nd strategy (TILs): not only TAAs, but also some mutations or further tumor specific antigens might be recognized; I would suggest to correct for “tumor antigens”

Reply 7: We agree and changed this term in Figure 2 to “resected tumor with T cells recognizing tumor antigens”.

Reviewer #1 comment 8: It would be nice to add the exact (or the range) of IL-2 doses instead of just “high” which is a bit vague (see lines 148, 173, 189); same comment for the number of infused NK92 cells (line 278).

Reply 8: We thank you for this important comment. However, we consider it too extensive to consistently provide the information on the exact dosage of cytokines and cell preparations for all specific cell therapy products / concepts. In our opinion, it is not relevant to state the precise dosage in the sections of text mentioned, so we have decided to only name the cytokines administered and to completely dispense the indication of the dosage. The same procedure was conducted for the suggested “number of infused NK92 cells”.

The following changes have been made:

  • Removal of “high doses” in line 164
  • Removal of “high doses” in line 180
  • Removal of “High-dose” in line 290
  • Removal of the number of infused NK92 cells in line 459.

Reviewer #1 comment 9: The quality of the figures is not very high, will this be improved for the final version?

Reply 9: We fully agree and will get in contact with our editor to improve figure quality and have the timeline (Figure 1) printed horizontally.

Reviewer 2 Report

In this manuscript, Wendel et al. give an historic overview on the use of NK and T cells as anti-tumor immunotherapeutic approaches. The authors have profusely reviewed the existing literature from the discovery of NK and T cells until latest advances in the field, such as redirection of effector cells by using chimeric antigen receptors (CAR).

The manuscript is overall well written and English is correct. Only minor spelling changes are required.

Minor comments:

- In Line 22 the s for “NK Cells” is missing

Line 35, "cells".

Line 36, I would write a comma instead of ¨or¨ for a better understanding.

Line 37, Allogeneic NK cells are also a possible off-the- self cell product.

-Figure 1 is too small; at its current size, the text is not legible. I would suggest to make this figure as full page and with horizontal orientation to be clearer for the reader.

Line 71, favor (to keep coherence with American English used throughout the manuscript).

-Line 146. As NK-92 cells have to be irradiated before being infused to patients, are there any publication exploring if irradiation impairs their in vivo proliferation/activation/cytotoxic ability?

Line 169, "cells".

Line 268, The name of the author is written in italics. Please correct.

-In line 371, I would suggest to cite this trial NCT04578210 which is exploring the safety Infusion of Natural Killer cells or Memory T Cells as Adoptive Therapy in COVID-19 pneumonia or lymphopenia.

Line 423, "demonstrated".

Line 484. Please note that this correction affects to Line 483. The sentence is too long. A comma or the word "these" between "trials" and "therapies" would improve the understanding of the sentence "Although CAR-T cells specific for CD19 showed high potential antitumor efficacy against relapsed and refractory B-ALL in clinical trials(,/these) therapies are limited to autologous settings even when they are HLA-matched... "

-References should be carefully reviewed and corrected:

      -Refs 30, 35, 36 (lines 639, 647 and 649) the name of the authors are written in capital letters.

               -Ref 38 is duplicated in 39 (lines 653 and 656)

      -Ref 78. Line 749. The name of the journal is written in capital letters

      -Ref 154. Line 965. The name of the journal is not correct.

      -A specific reference regarding EMA approval of Kymriah is missing in the references list

               -Ref 203. Line 1095. The name of the journal is missing.

-Please note that the name of some journals are written differently in some cases. E.g “Frontiers in Immunology” (Ref 199 and 213), or “New England Journal of Medicine” (Ref 174, 201).

Author Response

Reviewer #2 (Remarks to the Author):

Reviewer #2 comment 1: In this manuscript, Wendel et al. give an historic overview on the use of NK and T cells as anti-tumor immunotherapeutic approaches. The authors have profusely reviewed the existing literature from the discovery of NK and T cells until latest advances in the field, such as redirection of effector cells by using chimeric antigen receptors (CAR). 
The manuscript is overall well written and English is correct. Only minor spelling changes are required.

Minor comments from Reviewer #2:   

Reviewer #2 comment 2: In Line 22 the s for “NK Cells” is missing

Reply 2: We corrected this accordingly in line 22.          

Reviewer #2 comment 3: Line 35, "cells".

Reply 3: We changed this accordingly in line 35.

Reviewer #2 comment 4: Line 36, I would write a comma instead of ¨or¨ for a better understanding.

Reply 4: We thank the reviewer for this suggestion and changed this accordingly in line 36.

“[…], as a peripheral blood mononuclear cells (PBMCs) based therapeutic product, the adoptive transfer of specific T and NK cell products and the NK cell line NK-92.”

Reviewer #2 comment 5: Line 37, Allogeneic NK cells are also a possible off-the-shelf cell product.

Reply 5: We fully agree and corrected the respective sentence in the abstract as follows in line 37.

“Over the past decades, various immunotherapies have been developed, including cytokine-induced killer (CIK) cells, as a peripheral blood mononuclear cells (PBMCs) based therapeutic product, the adoptive transfer of specific T and NK cell products and the NK cell line NK-92. In addition to allogeneic NK cells, NK-92 cell products represent a possible “off-the-shelf” GMP-compliant therapeutic concept.”

Reviewer #2 comment 6: Figure 1 is too small; at its current size, the text is not legible. I would suggest to make this figure as full page and with horizontal orientation to be clearer for the reader.

Reply 6: We thank the reviewer for this important remark and will get in contact with our editor to improve figure quality and have the timeline (Figure 1) printed horizontally.

Reviewer #2 comment 7: Line 71, favor (to keep coherence with American English used throughout the manuscript).

Reply 7: We thank the reviewer for this suggestion and have changed “favour” to “favor” in line 72 for better coherence.

“The resulting imbalance in favor of activating signals is further complemented by the stress-associated upregulation of ligands for activating receptors, […].”

Reviewer #2 comment 8: Line 146. As NK-92 cells have to be irradiated before being infused to patients, are there any publication exploring if irradiation impairs their in vivo proliferation/activation/cytotoxic ability?

Reply 8: We thank the reviewer for this remark and have added the following paragraph in lines 149-152 to address this question.

“Furthermore, NK-92 cells must be irradiated prior to their application due to the lymphoma origin of this cell line, leading to a reduced in vivo proliferative capacity. The effect of irradiation on the cytotoxic capacity of NK-92 cells is controversially discussed [67,68].“

Reviewer #2 comment 9: Line 169, "cells".

Reply 9: We modified the sentence and implement the suggested term in line 174:

“In 1985, Steven Rosenberg and colleagues demonstrated the safe administration of LAK cells for patients with metastatic melanoma refractory to standard therapies in a phase I trial [76].”

Reviewer #2 comment 10: Line 268, The name of the author is written in italics. Please correct.

Reply 10: We changed this accordingly in line 281.

Reviewer #2 comment 11: In line 371, I would suggest to cite this trial NCT04578210 which is exploring the safety Infusion of Natural Killer cells or Memory T Cells as Adoptive Therapy in COVID-19 pneumonia or lymphopenia.

Reply 11: We thank the reviewer for this excellent suggestion to cite the RELEASE trial for our COVID-19 studies. We added the recommended trial in line 389.

Reviewer #2 comment 12: Line 423, "demonstrated".

Reply 12: We corrected this accordingly in line 441.

Reviewer #2 comment 13: Line 484. Please note that this correction affects to Line 483. The sentence is too long. A comma or the word "these" between "trials" and "therapies" would improve the understanding of the sentence "Although CAR-T cells specific for CD19 showed high potential antitumor efficacy against relapsed and refractory B-ALL in clinical trials(,/these) therapies are limited to autologous settings even when they are HLA-matched... "

Reply 13: We fully agree with the reviewer and adapted the paragraph in line 501-505 in favor for better readability as follows:

“Although CAR-T cells specific for CD19 showed high potential antitumor efficacy against relapsed and refractory B-ALL in clinical trials, these therapies are limited to autologous settings even when they are HLA-matched. Furthermore, generation of sufficient numbers of CAR-T cells for patients suffering from severe lymphopenia is impractical.”

Reviewer #2 comment 13.1: References should be carefully reviewed and corrected:    

Reply 13.1: Thank you for these precise proof reading of the whole manuscript, we added all the spelling changes to the revised version of the manuscript as indicated below.

Reviewer #2 comment 13.2: Refs 30, 35, 36 (lines 639, 647 and 649) the name of the authors are written in capital letters.      

Reply 13.2:
Upper-case letters of all Authors are corrected for all references.

Reviewer #2 comment 13.3: Ref 38 is duplicated in 39 (lines 653 and 656)        

Reply 13.3: Here we cited two different chapters that share a similar chapter title in the Ref. [38] and [39] (lines 764 and 767).

[38]      Mitchell, G.F.; Miller, J.F.A.P. Cell to Cell Interaction in the Immune Response. I. Hemolysin-Forming Cells in Neonatally Thymectomized Mice Reconstituted with Thymus or Thoracic Duct Lymphocytes. Journal of Experimental Medicine 1968, 128, 821–837, doi:10.1084/jem.128.4.821.

[39]      Mitchell, G.F.; Miller, J.F.A.P. Cell to Cell Interaction in the Immune Response. II. The Source of Hemolysin-Forming Cells in Irradiated Mice given Bone Marrow and Thymus or Thoracic Duct Lymphocytes. Journal of Experimental Medicine 1968, 128, 821–837, doi:10.1084/jem.128.4.821.

Reviewer #2 comment 13.4: Ref 78. Line 749. The name of the journal is written in capital letters

Reply 13.4: Upper-case letters of journals are all corrected.

Reviewer #2 comment 13.5: Ref 154. Line 965. The name of the journal is not correct.

Reply 13.5: The journal name has been corrected to “Clinical Cancer Research” (cited as “Clin. Cancer Res.”) for the updated Ref. [161] (line 1126).

Reviewer #2 comment 13.6:
A specific reference regarding EMA approval of Kymriah is missing in the references list

Reply 13.6: We thank the reviewer for this important remark and added the correct EMA reference describing the EMA approval of Kymriah in Ref [167] in line 1144.

[167]    Ali, S.; Kjeken, R.; Niederlaender, C.; Markey, G.; Saunders, T.S.; Opsata, M.; Moltu, K.; Bremnes, B.; Grønevik, E.; Muusse, M.; et al. The European Medicines Agency Review of Kymriah (Tisagenlecleucel) for the Treatment of Acute Lymphoblastic Leukemia and Diffuse Large B-Cell Lymphoma. The Oncologist 2020, 25, e321–e327, doi:https://doi.org/10.1634/theoncologist.2019-0233.

Reviewer #2 comment 13.7: Ref 203. Line 1095. The name of the journal is missing.

Reply 13.7: We thank the reviewer and added the journal name ”Haematologica” for Ref [210] in line 1265.

Reviewer #2 comment 13.8: Please note that the name of some journals are written differently in some cases. E.g “Frontiers in Immunology” (Ref 199 and 213), or “New England Journal of Medicine” (Ref 174, 201).

Reply 13.8: We agree that the journal names should be presented in a standardized format and corrected the references in the reference list. In accordance with the cancer guidelines, we adapted all journal names to the respective abbreviations.